# Novel Therapeutic Targets for the Treatment of Atopic Dermatitis

**DOI:** 10.3390/biomedicines11051303

**Published:** 2023-04-27

**Authors:** Gaku Tsuji, Kazuhiko Yamamura, Koji Kawamura, Makiko Kido-Nakahara, Takamichi Ito, Takeshi Nakahara

**Affiliations:** 1Research and Clinical Center for Yusho and Dioxin, Kyushu University Hospital, Fukuoka 812-8582, Japan; yamamura.kazuhiko.821@m.kyushu-u.ac.jp (K.Y.); nakahara.takeshi.930@m.kyushu-u.ac.jp (T.N.); 2Department of Dermatology, Graduate School of Medical Sciences, Kyushu University, Fukuoka 812-8582, Japan; kokawamu@gmail.com (K.K.); nakahara.makiko.107@m.kyushu-u.ac.jp (M.K.-N.); ito.takamichi.657@m.kyushu-u.ac.jp (T.I.)

**Keywords:** atopic dermatitis, biologic, small-molecule inhibitor

## Abstract

Atopic dermatitis (AD) is a chronic inflammatory skin disease that significantly impacts quality of life. The pathogenesis of AD is a complex combination of skin barrier dysfunction, type II immune response, and pruritus. Progress in the understanding of the immunological mechanisms of AD has led to the recognition of multiple novel therapeutic targets. For systemic therapy, new biologic agents that target IL-13, IL-22, IL-33, the IL-23/IL-17 axis, and OX40-OX40L are being developed. Binding of type II cytokines to their receptors activates Janus kinase (JAK) and its downstream signal, namely signal transduction and activator of transcription (STAT). JAK inhibitors block the activation of the JAK-STAT pathway, thereby blocking the signaling pathways mediated by type II cytokines. In addition to oral JAK inhibitors, histamine H4 receptor antagonists are under investigation as small-molecule compounds. For topical therapy, JAK inhibitors, aryl hydrocarbon receptor modulators, and phosphodiesterase-4 inhibitors are being approved. Microbiome modulation is also being examined for the treatment of AD. This review outlines current and future directions for novel therapies of AD that are currently being investigated in clinical trials, focusing on their mechanisms of action and efficacy. This supports the accumulation of data on advanced treatments for AD in the new era of precision medicine.

## 1. Introduction

Atopic dermatitis (AD) is a chronic inflammatory skin disease characterized by eczematous lesions with dry skin and intense pruritis [1,2]. A complex interplay of skin barrier dysfunction, type II inflammation, and pruritus underlies the pathogenesis of AD [1,2]. Epidermal disruption by external factors (allergens and pathogens) promotes the expression of interleukin (IL)-25, IL-33, and thymic stromal lymphopoietin (TSLP), which activate type II innate lymphoid cells (ILC2) and induce a type II immune response through the production of IL-5 and IL-13 [3]. In addition, TSLP-stimulated dendritic cells express OX40 ligand (OX40L), which binds to OX40 of T cells and induces the production of IL-4, IL-5, and IL-13 [4]. IL-4 and IL-13 cause skin barrier dysfunction, including decreased antibacterial peptides and filaggrin (FLG) and loricrin (LOR) expression, and further aggravate epidermal damage [5]. In addition to these Th2-deviated immune responses, the involvement of Th17 and Th22 cells as other immune axes in AD has also been shown [6] (Figure 1). Based on these findings, new biologics and small-molecule inhibitors are now being developed that target key molecules in the pathophysiology of AD. This review outlines novel therapeutic targets for the treatment of AD, including cytokines, the JAK (Janus kinase)-STAT (signal transduction and activator of transcription) pathway, OX40-OX40L interaction, histamine H4 receptor (H4R), aryl hydrocarbon receptor (AHR), phosphodiesterase (PDE) 4, and the microbiome.

## 2. Cytokines

### 2.1. IL-4/IL-13

It is widely recognized that abnormalities in type 2 immune response are central to AD. The Th2 cytokines IL-4 and IL-13 play major roles in the pathogenesis of this condition, which is supported by the high therapeutic efficacy of dupilumab, a monoclonal antibody that targets the receptor shared by IL-4 and IL-13 (IL-4Rα) [7]. At present, new antibodies CM310 (phase II, NCT04805411) and CBP201 (phase II, NCT05017480), which bind to a different region of IL-4Rα than that bound by dupilumab, are in development [8]. To reach further therapeutic endpoints of AD, these antibodies are designed to be administered less frequently and achieve a more rapid therapeutic effect than dupilumab. Previous studies have shown that IL-4 acts on Th0 cells to promote Th2 cell differentiation and proliferation among T cells and increases IgE production in B cells [1]. Furthermore, gene expression analysis on AD lesions has shown that IL-4 expression is almost undetectable in AD skin lesions, while IL-13 expression is predominant [9,10]. These findings suggest that IL-4 is more active in the central part of type 2 immune response, whereas IL-13 contributes more to local inflammation in the peripheral tissue, that is, the skin. It has also been shown that IL-13 expression in skin and blood correlates with the severity of AD [11,12].

Against the above background, antibody preparations targeting IL-13 alone are currently being developed. Tralokinumab is a fully human IgG4 antibody approved in the United States and Europe for the treatment of moderate to severe AD in adults. It binds to IL-13 and inhibits IL-13 signaling through IL-4Rα/IL13Rα1 and IL13Rα2 [13]. It is also expected to be approved in Japan. Lebrikizumab binds to IL-13 in a non-receptor binding domain distinct from that of tralokinumab and blocks IL-4Rα/IL13Rα1 heterodimerization and downstream signaling [14]. The difference is that tralokinumab inhibits IL-13 binding to IL-13Rα2 receptor, whereas lebrikizumab does not affect this binding. However, the function of IL-13Rα2 receptor remains largely unknown. Since the signaling motif of this receptor has not been identified, it is thought to function as a decoy receptor that regulates IL-13 levels [15]. The biological effects of inhibiting IL-13 binding to IL-13Rα2 receptor are currently unknown. Therefore, it is anticipated that the differences in the actions of tralokinumab and lebrikizumab will be revealed in future real-world clinical studies. In addition, comparing the treatment courses of dupilumab, tralokinumab, and lebrikizumab should lead to new insights into the roles of IL-4 and IL-13 in type 2 immune response in AD.

### 2.2. IL-22

In addition to Th2 cytokines, IL-22 has also been implicated in the pathogenesis of AD [1,6]. Under physiological conditions, IL-22 is produced in response to pathogen invasion [16]. IL-22 acts as an inflammatory cytokine and acts synergistically with IL-17 to induce β-defensins and antimicrobial peptides, such as S100A protein, in the epidermis [16]. IL-22 has also been shown to promote epidermal thickening and produce skin barrier defects [16]. Furthermore, the expression of IL-22 in mouse skin induces a strong Th2 immune response and suppresses epidermal differentiation [17]. In humans, blood IL-22 levels have been reported to correlate with the severity of AD [18].

In a phase II study with fezakinumab, an anti-IL-22 antibody, a statistically significant difference in the Severity Scoring of Atopic Dermatitis (SCORAD) scores was observed between the treatment and placebo groups at week 20 of treatment [19]. SCORAD is a clinical tool for assessing the severity of atopic dermatitis as objectively as possible [20]. SCORAD is calculated using the extent of disease spread and subjective symptoms, each accounting for around 20% of the total score, along with items related to disease intensity, which represents the remaining 60% [20]. In addition, in severe AD defined as SCORAD ≥ 50, a significantly stronger treatment effect was observed from week 6 of the study onward [19].

### 2.3. IL-31

IL-31, which is identified as a cytokine that directly stimulates nerves and causes pathological pruritis, is produced by T cells, basophils, eosinophils, mast cells, and macrophages in response to allergens and pathogens [21,22]. In addition, IL-31 receptors, namely IL-31 receptor alpha chain (IL-31RA) and oncostatin M receptor beta chain (OSMRβ), are expressed on macrophages, dendritic cells, eosinophils, basophils, epidermal cells, and cutaneous peripheral sensory nerves [21,22]. It has been reported that IL-31 administration to normal mice [23] or IL-31 transgenic mice induces scratching behavior [24], and conversely, anti-IL-31 mouse antibodies inhibit scratching behavior in mice with AD-like lesions [25]. Recent studies have also shown that IL-31 is involved not only in pruritis but also in epidermal thickening, impaired skin barrier function, and induction of inflammatory cytokines [26].

Nemolizumab is a humanized antibody that targets IL-31RA and specifically binds to IL-31 receptors on neurons, thereby inhibiting the action of IL-31 on nerves and improving pruritis. A phase III study in Japan showed its efficacy in the treatment of pruritus and skin lesions in AD [27]. Nemolizumab is also being investigated in prurigo nodularis [28], another condition characterized by chronic intense scratching. KPL-716 is an anti-oncostatin M receptor beta antibody that blocks IL-31 and oncostatin M signaling. KPL-716 has demonstrated good safety, tolerability, and efficacy against pruritus in a phase I study in patients with moderate to severe AD [29]. A phase II study on prurigo nodularis is also currently ongoing (NCT03816891).

### 2.4. IL-23/IL-17 Axis and IL-36

Although type II immune response is a central pathway in the pathogenesis of AD, recent studies have shown that the IL-23/IL-17 axis and IL-36, which are involved in the immune response against psoriasis, may also play a role in some AD phenotypes, including in Asian AD patients [1,6]. Clinical trials with secukinumab, an anti-IL-17A antibody, were conducted in patients with moderate to severe AD, but it did not show efficacy [30]. IL-23 is another candidate for the same psoriasis pathway, and a phase II trial of risankizumab on AD did not reach the primary endpoint of 75% reduction in eczema area and severity index (EASI) score (EASI-75) [31]. EASI is a validated scoring system that grades the physical signs of AD and is used for assessing the severity and extent of AD [32]. Spesolimab is a monoclonal antibody that targets IL-36 receptor and was developed for the treatment of generalized pustular psoriasis in adults [33]. A phase II study of spesolimab in adult patients with moderate to severe AD showed a reduction in EASI scores after 16 weeks in the treatment group [34].

### 2.5. IL-33

IL-33, an IL-1 family cytokine, is produced primarily in epithelial cells upon allergenic or microbial stimulation. IL-33 is involved in the induction of type II immune response by activating ILC2, which in turn produces large amounts of IL-5 and IL-13 [3]. IL-33 also decreases filaggrin (FLG) expression in keratinocytes, increases pruritus by acting on nerves, and activates Th1 and Th2 cells [3]. IL-33 is a driver of Th2-deviated inflammation and is upstream of IL-4 and IL-13. It has been shown that keratinocytes in AD lesional skin express higher levels of IL-33 compared to non-lesional skin of AD [35]. Furthermore, serum levels of IL-33 have been reported to be higher in AD patients than in healthy controls and correlated with disease severity [36]. These findings suggest that anti-IL-33 antibodies might be effective in the treatment of AD. However, phase II studies with anti-IL-33 antibodies (etokimab, REGN3500, and LY3375880) have not shown efficacy [37,38]. In addition, astegolimab, an antibody targeting ST2, the receptor for IL-33, was tested in a phase II study; however, astegolimab did not show a significant difference compared to placebo regarding the primary and secondary outcomes [39].

### 2.6. TSLP (Thymic Stromal Lymphopoietin)

Thymic stromal lymphopoietin (TSLP), one of the cytokines involved in the pathogenesis of AD, is produced in the epidermis by external and internal aggravating factors, such as pathogens, allergens, and Th2 cytokines. Then, through OX40L-OX40 binding, it promotes antigen-specific T-cell immune responses and differentiation into Th2 cells, and induces the production of IL-4, IL-5, and IL-13 [4]. Furthermore, IL-4 and IL-13 induce the production of TSLP. Thus, epidermis-derived TSLP acts as an initiator of inflammation in AD and as a master switch for the atopic march [40]. These findings have led to the expectation that anti-TSLP antibodies would have high therapeutic efficacy against AD, but efficacy has not yet been confirmed. Tezepelumab was developed as a human anti-TSLP monoclonal antibody, but clinical trials failed to demonstrate its pre-determined efficacy [41] and were terminated. Both CM326 and BSI-045B are anti-TSLP antibodies in early development. CM326 (phase I/II, NCT05186922) and BSI-045B (phase I, NCT05114889) have been designed for a higher biological activity than tezepelumab and are expected to achieve better results.

## 3. Janus Kinase (JAK)—Signal Transduction and Activator of Transcription (STAT) Pathway

Despite the recent approval of dupilumab, there remains an unmet need for a treatment that provides more efficacious results for patients with moderate to severe AD. Binding of type II cytokines to their receptors activates Janus kinase (JAK) and its downstream signal, namely signal transduction and activator of transcription (STAT) [42]. Therefore, the JAK-STAT pathway is an important cytokine signaling molecule involved in the pathogenesis of AD [42]. In contrast to biologics, JAK inhibitors can block multiple cytokine signaling pathways simultaneously, thus exerting broader immunomodulatory activity [42]. The pathogenesis of AD is thought to be complex and involves signaling pathways through four JAKs (JAK1, JAK2, JAK3, and tyrosine kinase 2 (TYK2)) [42,43]. Among them, JAK1 is of particular interest because it acts downstream of cytokines, such as IL-4, IL-13, IL-31, and TSLP, which are key cytokines in the pathogenesis of AD [42,43]. JAK inhibitors approved to date include pan-JAK inhibitors (delgocitinib), JAK1/2 inhibitors (baricitinib and ruxolitinib), and selective JAK1 inhibitors (upadacitinib and abrocitinib) [42,43].

Upadacitinib, abrocitinib, and baricitinib are oral JAK inhibitors approved for AD; each has been shown to achieve EASI-75 within 16 weeks of administration as monotherapy or in combination with topical corticosteroids in a significantly higher proportion of patients than in the placebo group, as well as rapid improvement of pruritis [44,45,46,47,48,49,50,51]. Although there are no reports on baricitinib, upadacitinib and abrocitinib have been studied in head-to-head clinical trials with dupilumab in order to clarify their differences in efficacy [44,49]. In a clinical study involving 30 mg of upadacitinib daily versus 300 mg of dupilumab once every 2 weeks in patients with moderate to severe AD, upadacitinib showed superior efficacy. Upadacitinib achieved EASI-75 as early as week 2 and significantly higher pruritis improvement rates as early as week 1. Treatment effects, such as improvements in skin lesions and pruritis, were realized earlier with upadacitinib than with dupilumab [44].

In a phase 3 study, patients with moderate to severe AD received either 200 or 100 mg of abrocitinib orally once daily, 300 mg of dupilumab subcutaneously once every 2 weeks, or placebo with topical therapy (including topical corticosteroids, topical calcineurin inhibitors, or topical phosphodiesterase-4 inhibitors) for 16 weeks. The EASI-75 rate by week 12 was significantly higher in the 200 mg of abrocitinib and 100 mg of abrocitinib groups than in the placebo group. Abrocitinib at 200 mg was superior to dupilumab in terms of improving pruritis at week 2 [49]. Upadacitinib and abrocitinib suppress pruritis earlier than dupilumab does by simultaneously inhibiting the JAK-STAT pathway involving multiple inflammatory mediators associated with pruritis, including IL-4, IL-13, IL-31, and TSLP.

The JAK-STAT pathway mediates both inflammation and normal physiological functions, such as cell proliferation and hematopoietic function [52]. Therefore, adverse event (AE) monitoring is important in treatment with oral JAK inhibitors. The major AEs associated with oral JAK inhibitors in AD include gastrointestinal disturbances, nasopharyngitis, and headache, which were reported to occur more frequently than in a placebo group [53]. Meanwhile, no cases of malignancy or death were reported [53]. Elevated creatine phosphokinase levels were noted with baricitinib and upadacitinib treatments. Patients with elevated creatine phosphokinase levels were asymptomatic [53], but further investigation is needed. Serious infections, herpes zoster, and acne were reported to be more common in patients treated with JAK inhibitors, but their frequencies were still very low [44,48,50].

Topical JAK inhibitors have been developed, which act locally at the lesion and are, thus, associated with fewer AEs than oral JAK inhibitors. The first topical JAK inhibitor, ruxolitinib cream, was approved by the FDA in 2021 and is available in the US [54]. Meanwhile, in Japan, delgocitinib ointment can be used for both pediatric and adult AD patients [55,56]. In addition, tofacitinib (phase II) [57] and cerdulatinib (phase I) [58] are under development.

## 4. OX40-OX40L Interaction

OX40 (CD134) is a co-stimulatory molecule belonging to the TNF receptor superfamily that is expressed on T cells and binds to OX40L (CD252), which is expressed on dendritic cells and other antigen-presenting cells [4]. OX40 expression is increased by antigen-specific T-cell receptor activation [4]; therefore, inhibition of OX40 may be able to intensively suppress specific T cells involved in AD rather than achieve systemic immunosuppression. However, because the OX40-OX40L-mediated antigen presentation process is also an important pathway in tumor immunity that protects against tumorigenesis [59], the impact of drugs that inhibit OX40-OX40L on tumor development is being carefully examined in terms of long-term safety.

Telazorlimab is a humanized antibody against OX40. In a phase II study, AD patients treated with telazorlimab demonstrated suppression of Th2, Th1, and Th17/Th22 cytokines in the treatment arm with reductions in OX40+ T cells and OX40L+ dendritic cells. In addition, the primary endpoint of EASI-75 at week 16 was achieved at a higher rate in the treatment group than in the placebo group [60].

Rocatinlimab is another humanized antibody against OX40. A phase II study showed that it achieved statistically significant improvement in EASI scores at week 16 versus placebo at all doses; the rate of achievement of EASI-75 at week 16 was significantly higher in the treatment group. Furthermore, the achievement of EASI-75 was maintained during the drug-free follow-up period [61]. These results provide interesting evidence that rocatinlimab maintains therapeutic efficacy long after discontinuation.

KY1005 is an anti-OX40L antibody. In a phase II clinical trial (NCT03754309), preliminary results showed that the group treated with KY1005 showed significant improvement in EASI scores at 16 weeks compared to the scores achieved by the placebo group [62].

## 5. Histamine H4 Receptor (H4R)

Four histamine receptor types (H1R–H4R) have been identified, all of which are G-protein-coupled receptors [63]. Histamine H4 receptor (H4R) is expressed on most of the cells involved in the pathogenesis of AD, including T cells, Langerhans cells, dendritic cells, macrophages, mast cells, eosinophils, basophils, keratinocytes, fibroblasts, and neurons [64]. In addition to direct histamine-induced itch, H4R is associated with many immune responses, including histamine-induced chemotaxis of immune cells, expression of adhesion molecules, and regulation of cytokine and chemokine production [65]. Therefore, H4R antagonists may be promising candidates for the treatment of AD by reducing pruritis and suppressing inflammatory responses. JNJ-39758979, ZPL-3893787, and LEO152020 have been developed as H4R antagonists.

In a phase II study, JNJ-39758979 was administered for six weeks to adult Japanese patients with moderate AD. Significant reduction in pruritis and improvement in skin lesions were observed in the treatment group compared to the findings in the placebo group. However, this study was terminated due to the development of agranulocytosis, which appeared to be metabolite-related rather than due to H4R antagonism [66]. In addition, ZPL-3893787 was administered for eight weeks to adult patients with moderate to severe AD in a phase II study. ZPL-3893787 was well tolerated overall. Significant reductions in EASI and SCORAD scores were observed in the treatment group compared to the findings in the placebo group. However, in terms of the effect on pruritus, no significant improvement was shown [67]. This led to the discontinuation of the development of ZPL-3893787. Meanwhile, LEO 152020 was developed for the treatment of chronic urticaria and AD. A phase II study is currently underway to evaluate its efficacy and safety upon administration for 16 weeks (NCT05117060).

## 6. Aryl Hydrocarbon Receptor (AHR)

Aryl hydrocarbon receptor (AHR) is a chemical receptor, particularly highly expressed in the skin, that is involved in keratinocyte differentiation and proliferation, inflammatory cytokine production, and immune regulation of Th17/22 and regulatory T cells [68]. Recently, both beneficial and deleterious effects of AHR signaling in inflammatory skin diseases, particularly atopic dermatitis, have been demonstrated [69].

Filaggrin (FLG) is a protein important for epidermal differentiation and stratum corneum function [70]. Loss-of-function mutations in FLG have been observed in some AD patients [71,72] and affect physical skin barrier, leading to antigen penetration into the subepidermal layer and activation of immune responses. AD patients with FLG mutations have been reported to have more persistent and more severe symptoms of AD, to be more likely to be accompanied by allergic sensitization, and to have more significant deficiencies in natural moisturizing factors [73]. In addition to FLG, loricrin (LOR) and involucrin (IVL) dysfunctions have been shown to be involved in the pathogenesis of AD [74]. All of these proteins are encoded by genes of the epidermal differentiation complex (EDC) located on chromosome 1q21.3 [75]. Several studies have shown that activation of AHR is key to FLG expression in human keratinocytes [76,77,78]. Binding of endogenous and exogenous ligands to AHR induces AHR translocation from the cytoplasm to the nucleus. The activated AHR binds to the EDC locus, leading to the upregulation of FLG, LOR, IVL, and other barrier-related proteins [79,80,81]. Furthermore, activation of AhR upregulates OVOL1 [82], which has been shown to be an important transcription factor for the expression of EDC genes, including FLG and LOR [77,78,80]. In AD, IL-4 inhibits the nuclear translocation of OVOL1, which results in the downregulation of FLG and LOR [78]. AHR activation by AHR ligands, such as FICZ, induces the nuclear translocation of OVOL1, which reverses the downregulation of FLG and LOR [78]. Moreover, AHR regulates the activation of NRF2 (nuclear factor erythroid 2-related factor 2), a transcription factor that induces the expression of a series of cytoprotective genes encoding detoxification and antioxidant enzymes [83,84,85]. Activation of NRF2 reportedly interferes with Th2 cytokine signaling through the dephosphorylation of STAT6 [81]. NRF2 also suppresses the production of pro-inflammatory cytokines, such as IL-6 and IL-1β, via the impairment of NF-κB transcriptional activity [86].

AHR also affects pruritus in AD. The ARTN gene encodes the neurotrophic factor artemin, which is responsible for pruritus. ARTN is specific to keratinocytes and is a target gene of AHR [87,88]. The expression pattern of artemin upon AHR activation depends on the AHR ligand. The endogenous AHR ligand FICZ does not increase artemin expression [87]. On the other hand, AHR ligands such as diesel exhaust particles (DEPs) and their major constituent, DMBA (7,12-dimethylbenz[a]anthracene), increase artemin expression [87]. The difference is inferred to be related to FICZ being metabolized and inactivated before the expression of AD-related AhR target genes is induced. In contrast, non-degradable ligands, such as DMBA, are retained in the skin, resulting in prolonged AhR activation [87]. Thus, prolonged activation of AHR by air pollutants is necessary to cause pruritus in AD [87]. Indeed, several epidemiological studies have demonstrated that exposure of skin to air pollutants acts as a risk factor for the development or exacerbation of AD [89,90]. Air pollutants include particulate matter (PM), a mixture of airborne solid or liquid particles. Outdoor concentrations of PM have been shown to be associated with increased AD severity and pruritis [91,92]. PM contains the AHR ligands PAHs, which induce AHR-dependent CYP1A1mRNA [87,93]. In addition to PM, exposure to tobacco, which also contains PAHs, is associated with a higher prevalence of AD in both children and adults [94]. It has also been reported that activation of AHR in the lesional skin of AD patients is positively correlated with ARTN expression, which is associated with allokinesis, epidermal hyperinnervation, and inflammation [87]. These reports support the hypothesis that activation of AHR is involved in the development of AD caused by air pollutants.

Meanwhile, when chemicals classified as therapeutic AHR-modulating agents (TAMAs) [95] act on AHR, the disease activity of AD is suppressed via increases in the levels of skin barrier proteins (such as FLG and LOR) and the induction of NRF2. Tapinarof, a drug categorized as a TAMA, was developed as a topical treatment for both psoriasis and AD [96,97]. Tapinarof has been identified as a dual activator of AHR and NRF2 in human keratinocytes [98]. Although it has been shown that tapinarof upregulates FLG and LOR in human keratinocytes [99], whether it affects artemin expression has not been reported. Tapinarof cream was approved for use in psoriasis by the FDA in May 2022 after its efficacy was confirmed in a clinical trial [96]. A phase II trial in adults with AD showed that tapinarof achieved significant improvements in EASI scores and pruritis after 12 weeks of topical application [97]. A phase III trial for AD is currently underway in Europe, the US, and Japan. A phase II study in pediatric AD patients is also underway (NCT05014568).

## 7. Phosphodiesterase 4 (PDE4)

Phosphodiesterase 4 (PDE4) is an intracellular enzyme that degrades cyclic adenosine monophosphate (cAMP). Therefore, PDE4 inhibitors are expected to decrease inflammatory cytokines by increasing cAMP levels [100]. While apremilast, an oral PDE4 inhibitor, is already approved for the treatment of psoriasis, its development for AD has not progressed [101]. A phase II trial of apremilast in AD showed significant improvement only at higher doses compared to placebo, but the trial was terminated due to adverse events [101].

The development of topical PDE4 inhibitors, which are thought to be associated with a lower frequency of adverse events (gastrointestinal disturbances and headache) observed upon systemic administration, is underway. Crisaborole is the first topical PDE4 inhibitor developed for AD. A phase III study in patients with mild/moderate AD aged two years and older showed an improvement in clinical signs in the crisaborole group [102]. Crisaborole is now being used in patients with mild to moderate AD after being approved by the FDA for use in patients aged three months and older, but not outside the US. In Japan, investigation on difamilast was completed in a phase III study in children and adults, and it has been approved for patients aged two years and older [103]. In addition, investigation on roflumilast was completed in a phase III study in psoriasis and showed favorable results compared to the findings in the control group [104]. However, for AD, a phase II study showed no statistically significant difference in the absolute change in EASI scores at week 4, the primary endpoint, which was due to the small number of eligible patients [105]. Although there was no significance in the primary endpoint, based on the favorable safety profile and promising results, a phase III trial is currently underway (NCT04845620).

## 8. Microbiome

The microbiome of the skin has recently been shown to play an important role in the etiology of AD, as the microbiome functions as a regulator of innate and acquired immunity [106,107]. Previous studies have shown that the skin of AD patients is characterized not only by a low microbial diversity but also by a high colony-formation rate of *Staphylococcus aureus* and reduced indigenous bacteria (*Streptococcus*, *Corynebacterium*, *Cutibacterium*, and Proteobacteria) [106,107]. It has also been suggested that the diversity of the microbiome may correlate with the severity of AD [108]. Therefore, strategies to treat AD by regulating the skin microbiome (correcting dysbiosis) and replacing specific microorganisms are being investigated. Oral microbiome regulators are under development, with several being tested in phase I trials. The following describes these topical formulations.

*Staphylococcus hominis A9* (ShA9), a bacterium isolated from healthy human skin, has been shown in animal studies to have two activities: killing *Staphylococcus aureus* and inhibiting the production of *Staphylococcus aureus*-derived toxins [109]. In a phase I/II study, ShA9 was applied twice daily for seven days in patients with moderate to severe AD, who were tested positive for *Staphylococcus aureus* colony formation. The results showed that ShA9 reduced *Staphylococcus aureus* colony formation and improved EASI and SCORAD scores [109].

*Nitrosomonas eutropha* (B244) is a bacterium that produces nitric oxide [69]. Nitric oxide is an important mediator with potential anti-inflammatory effects and, thus, has therapeutic potential for AD [110]. In a phase II study in adults with AD, B244 administered as a spray significantly improved pruritis (NCT03775434). A phase II study on B244 is also currently underway (NCT04490109).

## 9. Conclusions

The number of novel agents for the treatment of AD is expected to increase as the pathophysiology of this condition becomes clearer (Table 1). Elucidation of the endotypes of AD, of which there are various phenotypes, and their linkage to precision medicine is a major challenge for the future. To achieve this goal, it is important to establish a high-quality biorepository consisting of skin biopsy tissues, blood samples, non-invasive tape strip specimens from the skin, and skin swab specimens to assess the microbiome, as well as a technology platform for their integrated analysis. Further progress on the identification of predictive and prognostic biomarkers for companion diagnostics will also be needed.

Tralokinumab and lebrikizumab are monoclonal antibodies that antagonize IL-13. CM310 and CBP201 are antibodies binding to a different region of IL-4Rα than that bound by dupilumab. Fezakinumab is a monoclonal antibody against interleukin-22. Nemolizumab is an anti-IL-31 receptor A monoclonal antibody approved in Japan. KPL-716 is an anti-oncostatin M receptor (OSMR) beta antibody that blocks IL-31 and oncostatin M signaling. Spesolimab is an IL-36 receptor antibody for the treatment of generalized pustular psoriasis. CM326 and BSI-045B are monoclonal antibodies that target thymic stromal lymphopoietin (TSLP). Delgocitinib, ruxocitinib, tofacitinib, and cerdulatinib are topical JAK inhibitors. These JAK inhibitors prevent the activation of signal transduction and activator of transcription (STAT), a downstream signal of JAK. Telazorlimab and rocatinlimab are antibodies against OX40. KY1005 is an antibody against OX40L. LEO152020 is an antagonist of histamine H4 receptor (H4R). Tapinarof is a therapeutic aryl hydrocarbon receptor (AHR)-modulating agent that is utilized as a topical cream. Crisaborole, difamilast, and roflumilast are topical agents of phosphodiesterase (PDE)-4 inhibitor. ShA9 and B244 are bacteria that correct the dysbiosis of AD.

## Figures and Tables

**Figure 1 biomedicines-11-01303-f001:**
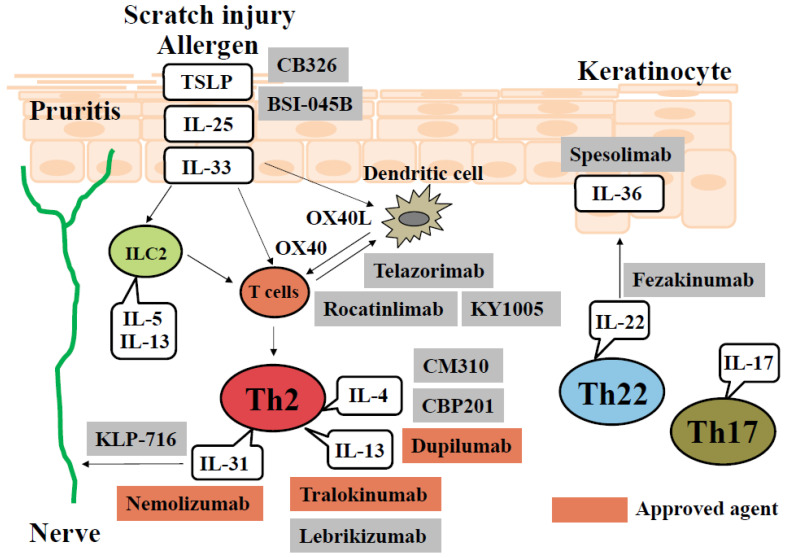
Novel therapeutic biologic targets in the pathogenesis of AD. Scratch injury induced by pruritis and allergens produce TSLP, IL-25, and IL-33, which are cytokines classified as alarmins, in keratinocytes. When these act on ILC2, they induce the production of IL-5 and IL-13, resulting in differentiation and proliferation that is prone toward Th2. This pathway is enhanced via OX40L-OX40 interaction between dendritic cells and T cells. Th2-cell-derived IL-4 and IL-13 produce skin barrier dysfunction in keratinocytes, which enhances scratch injury-induced and allergen-induced cell damage. In addition to the Th2 immune response, IL-17 from Th17 cells and IL-22 from Th22 cells induce acanthosis in the epidermis, which contributes to the formation of intractable lichenified AD skin lesions. CB326 and BSI-045: anti-TSLP Ab. Telazorimab and rocatinlimab: anti-OX40 Ab. KY1005: anti-OX40L antibody. CM310, CBP201, and dupilumab: anti-IL4Rα antibody. Tralokinumab and lebrikizumab: anti-IL-13 Ab. Nemolizumab: anti-IL-31 Ab. KLP-716: anti-oncostatin M receptorβ Ab. Spesolimab: anti-IL-36 receptor Ab. Fezakinumab: anti-IL-22 Ab.

**Table 1 biomedicines-11-01303-t001:** Development of agents for AD under investigation in clinical trials.

	Phase I/II	Phase III	Approved
IL-4/IL-13	CM310 CBP201	Lebrikizumab	* Tralokinumab
IL-22	Fezakinumab		
IL-31RA/OSMRβ	KPL-716		** Nemolizumab
IL-23·IL-36	Spesolimab		
TSLP	CM326 BSI-045B		
JAK-STAT	Tofacitinib Cerdulatinib		** Delgocitinib *** Ruxocitinib
OX40-OX40L	Telazorlimab Rocatinlimab KY1005		
H4R	LEO152020		
AHR		Tapinarof	
PDE4		Roflumilast	*** Crisaborole * Difamilast
Microbiome	ShA9 B244		
			* USA and Europe only
			** Japan only
			*** USA only

* Agents approved in USA and Europe only. ** Agents approved in Japan only. *** Agents approved in USA only.

## Data Availability

Not applicable.

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
