# Peer review of "Novel Therapeutic Targets for the Treatment of Atopic Dermatitis"

_biomedicines, 2023, doi:10.3390/biomedicines11051303_

Round 1

Reviewer 1 Report

Very interesting and well-written article, reflects the special issue where it was submitted.

All the new targets for atopic dermatitis have been selected. 

I believe that some paragraphs in the article contain serious errors and need to be further investigated given the future importance

1) The AHR section, only one reference was identified, there are many in the literature that extensively explain the role of AHR in Atopic Dermatitis, I recommend the authors to add the references I list here and elaborate on the topic

- doi:10.3390/cells10123559

- doi:10.3390/antiox11020227

2) The JAK stat paragraph contains many errors, the approval of UPADACITINIB in atopic dermatitis is not mentioned, the importance of this drug in the pathology is crucial, case reports and studies are reported in the literature, I advise the authors to cite the references I list here expanding this paragraph a lot because I think it is very useful for the literature

- DOI:10.1001/jamadermatol.2021.3023

- doi:10.1016/S0140-6736(21)00589-4

- doi:10.1111/jdv.18137

- doi:10.1016/S0140-6736(21)00588-2

3) Abrocitinib should be mentioned in the JAK-STAT section.

4) The table should be updated, upadacitinib has approval it is useless to mention the phase 3 study that has been over for a long time, same for abrocitinib, the authors need to study better this topic

5)Minor editing of English language required

Minor editing of English language required

Author Response

Reviewer #1

1) The AHR section, only one reference was identified, there are many in the literature that extensively explain the role of AHR in Atopic Dermatitis, I recommend the authors to add the references I list here and elaborate on the topic

- doi:10.3390/cells10123559

- doi:10.3390/antiox11020227

Thank you very much for the comment. We cited the recommended two articles in the manuscript. As you pointed out, the role of AHR in atopic dermatitis has been reported to be both beneficial and detrimental. Based on our report and the findings of other researchers, we have described the mechanisms of each and amended the part on aryl hydrocarbon receptor (AHR) as follows:

(Lines 290–351)

Recently, both beneficial and deleterious effects of AHR signaling in inflammatory skin diseases, particularly atopic dermatitis, have been demonstrated [69].

Filaggrin (FLG) is a protein important for epidermal differentiation and stratum corneum function [70]. Loss-of-function mutations in FLG have been observed in some AD patients [71,72] and affect the physical skin barrier, leading to antigen penetration into the subepidermal layer and activation of immune responses. AD patients with FLG mutations have been reported to have more persistent and more severe symptom of AD, to be more likely to be accompanied by allergic sensitization, and to have more significant deficiencies in natural moisturizing factors [73]. In addition to FLG, loricrin (LOR) and involucrin (IVL) dysfunctions have been shown to be involved in the pathogenesis of AD [74]. All of these proteins are encoded by genes of the epidermal differentiation complex (EDC) located on chromosome 1q21.3 [75]. Several studies have shown that activation of AHR is key to FLG expression in human keratinocytes [76–78]. Binding of endogenous and exogenous ligands to AHR induces AHR translocation from the cytoplasm to the nucleus. The activated AHR binds to the EDC locus, leading to upregulation of FLG, LOR, IVL, and other barrier-related proteins [79-81]. Furthermore, activation of AhR upregulates OVOL1 [82], which has been shown to be an important transcription factor for the expression of EDC genes including FLG and LOR [77,78,80]. In AD, IL-4 inhibits the nuclear translocation of OVOL1, which results in the downregulation of FLG and LOR [78]. AHR activation by AHR ligands such as FICZ induces the nuclear translocation of OVOL1, which reverses the downregulation of FLG and LOR [78]. Moreover, AHR regulates the activation of NRF2 (nuclear factor erythroid 2-related factor 2), a transcription factor that induces the expression of a series of cytoprotective genes encoding detoxification and antioxidant enzymes [83–85]. Activation of NRF2 reportedly interferes with Th2 cytokine signaling through the dephosphorylation of STAT6 [81]. NRF2 also suppresses the production of pro-inflammatory cytokines such as IL-6 and IL-1β via the impairment of NF-κB transcriptional activity [86].

AHR also affects pruritus in AD. The ARTN gene encodes the neurotrophic factor artemin, which is responsible for pruritus. ARTN is specific to keratinocytes and a target gene of the AHR [87,88]. The expression pattern of artemin upon AHR activation depends on the AHR ligand. The endogenous AHR ligand FICZ does not increase artemin expression [87]. On the other hand, AHR ligands such as diesel exhaust particles (DEPs) and their major constituent, DMBA (7,12-dimethylbenz[a]anthracene), increase artemin expression [87]. The difference is inferred to be that FICZ is metabolized and inactivated before the expression of AD-related AhR target genes is induced. In contrast, non-degradable ligands such as DMBA are retained in the skin, resulting in prolonged AhR activation [87]. Thus, prolonged activation of the AHR by air pollutants is necessary to cause pruritus in AD [87]. Indeed, several epidemiological studies have demonstrated that the exposure of skin to air pollutants acts as a risk factor for the development or exacerbation of AD [89,90]. Air pollutants include particulate matter (PM), a mixture of airborne solid or liquid particles. Outdoor concentrations of PM have been shown to be associated with increased AD severity and pruritis [91,92]. PM contains the AHR ligands PAHs, which induce AHR-dependent CYP1A1mRNA [87,93]. In addition to PM, exposure to tobacco, which also contains PAHs, is associated with a higher prevalence of AD in both children and adults [94]. It was also reported that activation of AHR in the lesional skin of AD patients is positively correlated with ARTN expression, which is associated with allokinesis, epidermal hyperinnervation, and inflammation [87]. These reports support the hypothesis that activation of AHR is involved in the development of AD caused by air pollutants. 

Meanwhile, when chemicals classified as therapeutic AHR-modulating agents (TAMAs) [95] act on AHR, the disease activity of AD is suppressed via increases in the levels of skin barrier proteins (such as FLG and LOR) and the induction of NRF2. Tapinarof, a drug categorized as a TAMA, was developed as a topical treatment for both psoriasis and AD [96,97]. Tapinarof has been identified as a dual activator of AHR and NRF2 in human keratinocytes [98]. Although it has been shown that tapinarof upregulates FLG and LOR in human keratinocytes [99], whether it affects artemin expression has not been reported. Tapinarof cream was approved for use in psoriasis by the FDA in May 2022 after its efficacy was confirmed in a clinical trial [96]. A Phase II trial in adults with AD showed that tapinarof achieved significant improvement of EASI scores and pruritis after 12 weeks of topical application [97].

2) The JAK stat paragraph contains many errors, the approval of UPADACITINIB in atopic dermatitis is not mentioned, the importance of this drug in the pathology is crucial, case reports and studies are reported in the literature, I advise the authors to cite the references I list here expanding this paragraph a lot because I think it is very useful for the literature.

3) Abrocitinib should be mentioned in the JAK-STAT section.

- DOI:10.1001/jamadermatol.2021.3023

- doi:10.1016/S0140-6736(21)00589-4

- doi:10.1111/jdv.18137

- doi:10.1016/S0140-6736(21)00588-2

Thank you very much for the comment. We cited the recommended four articles. We described the importance of the JAK-STAT pathway in atopic dermatitis and the efficacy of its inhibitors, upadacitinib, abrocitinib and baricitinib. In addition, we described the therapeutic efficacy of upadacitinib and abrocitinib, focusing on comparison of their efficacy with that of dupilumab. Based on this, we amended the part on the Janus kinase (JAK)-signal transduction and activator of transcription (STAT) pathway as follows:

(Lines 179–234)

Janus kinase (JAK)-signal transduction and activator of transcription (STAT) pathway

Despite the recent approval of dupilumab, there remains an unmet need for a treatment that provides more efficacious results for patients with moderate to severe AD. Binding of type II cytokines to their receptors activates Janus kinase (JAK) and its downstream signal, signal transduction and activator of transcription (STAT) [42]. Therefore, the JAK-STAT pathway is an important cytokine signaling molecule involved in the pathogenesis of AD [42]. In contrast to biologics, JAK inhibitors can block multiple cytokine signaling pathways simultaneously, thus exerting broader immunomodulatory activity [42]. The pathogenesis of AD is thought to be complex and involves signaling pathways through four JAKs (JAK1, JAK2, JAK3, and Tyrosine Kinase 2: TYK2) [42,43]. Among them, JAK1 is of particular interest because it acts downstream of cytokines such as IL-4, IL-13, IL-31, and TSLP, which are key cytokines in the pathogenesis of AD [42,43]. JAK inhibitors approved to date include pan-JAK inhibitors (delgocitinib), JAK1/2 inhibitors (baricitinib, ruxolitinib), and selective JAK1 inhibitors (upadacitinib, abrocitinib) [42,43].

Upadacitinib, abrocitinib, and baricitinib, oral JAK inhibitors approved for AD, each achieved EASI-75 within 16 weeks of administration as mono-therapy or in combination with topical corticosteroids in a significantly higher proportion of patients than in the placebo group, and achieved rapid improvement of pruritis [44–51]. Although there are no reports on baricitinib, upadacitinib and abrocitinib have been studied in head-to-head clinical trials with dupilumab in order to clarify differences in efficacy [44,49]. In a clinical study of 30 mg of upadacitinib daily versus 300 mg of dupilumab once every 2 weeks in patients with moderate to severe AD, upadacitinib showed superior efficacy. Upadacitinib achieved EASI-75 as early as week 2 and significantly higher pruritis improvement rates as early as week 1. Treatment effects such as improvement in skin lesions and pruritis were realized earlier with upadacitinib than with dupilumab [44].

In a phase III study, patients with moderate to severe AD received either 200 or 100 mg of abrocitinib orally once daily, 300 mg of dupilumab subcutaneously once every 2 weeks, or placebo with topical therapy (including topical corticosteroids, topical calcineurin inhibitors, or topical phosphodiesterase-4 inhibitors) for 16 weeks. The EASI-75 rate by week 12 was significantly higher in the 200 mg abrocitinib and 100 mg abrocitinib groups than in the placebo group. Abrocitinib at 200 mg was superior to dupilumab in terms of improving pruritis at week 2 [49]. Upadacitinib and abrocitinib suppress pruritis earlier than dupilumab does by simultaneously inhibiting the JAK-STAT pathway of multiple inflammatory mediators associated with pruritis, including IL-4, IL-13, IL-31, and TSLP.

The JAK-STAT pathway mediates both inflammation and normal physiological functions such as cell proliferation and hematopoietic function [52]. Therefore, adverse event (AE) monitoring is important in treatment with oral JAK inhibitors. The major AEs of oral JAK inhibitors in AD include gastrointestinal disturbances, nasopharyngitis, and headache, which were reported to occur more frequently than in a placebo group [53]. Meanwhile, no cases of malignancy or death were reported [53]. Elevated creatine phosphokinase levels were noted with baricitinib and upadacitinib treatments. Patients with elevated creatine phosphokinase levels were asymptomatic [53], but further investigation is needed. Serious infections, herpes zoster, and acne have been reported to be more common in patients treated with JAK inhibitors, but their frequencies are still very low [44,48,50].

Topical JAK inhibitors have been developed, which act locally at the lesion and are thus associated with fewer AEs than oral JAK inhibitors. The first topical JAK inhibitor, ruxolitinib cream, was approved by the FDA in 2021 and is available in the US [54]. Meanwhile, in Japan, delgocitinib ointment can be used for both pediatric and adult AD [55, 56]. In addition, tofacitinib (Phase II) [57] and cerdulatinib (Phase I) [58] are under development.

4) The table should be updated, upadacitinib has approval it is useless to mention the phase 3 study that has been over for a long time, same for abrocitinib, the authors need to study better this topic

Thank you very much for the comment. We deleted upadacitinib, abrocitinib, and baricitinib in Table 1. Instead, we have added tofacitinib and cerdulatinib, which are in development as topical agents, to the Phase I/II portion. We have also added them to the manuscript accordingly.

(Lines 324–325)

In addition, tofacitinib (Phase II) [57] and cerdulatinib (Phase I) [58] are under development.

5)Minor editing of English language required

Thank you very much for the comment. The manuscript has been edited by an English proofreading service.

Reviewer 2 Report

This is a very comprehensive overview of emerging targets for treating atopic dermatitis. In this era of continuous development, such an overview is very useful.

Please make sure that all abbreviations are written in full or explained when given for the first time, even if very known and logical. This is usually demanded by all journals, but also please have in mind who you want the paper to be understood by:

What is the target readership of the paper? A clinical dermatologist would easily understand SCORAD, EASI, EASI-75 by week X, etc. and what reduction of these means, but how much do those abbreviations mean to an immunologist or biomedical? On the other hand, many immunological abbreviations may not be readily understood by a clinical dermatologist, although clear for i.e. immunologists.

Tables should also be explained in full and be understood without having to refer to the text. All abbreviations used in tables also need to be written in full as table footnotes or legend.

Suggestion: It may also be of interest to mention potential side effects when targeting a certain pathway.

Not explained in full when first mentioned (both abstract, text and table/figures): JAK, PDE4, TSLP, JAK-STAT, SCORAD, EASI, EASI-75, TYK...

Author Response

Reviewer #2

Please make sure that all abbreviations are written in full or explained when given for the first time, even if very known and logical. This is usually demanded by all journals, but also please have in mind who you want the paper to be understood by.

Thank you very much for the comment. We have added full spellings for the first occurrence of abbreviations to aid understanding by the reader.

What is the target readership of the paper? A clinical dermatologist would easily understand SCORAD, EASI, EASI-75 by week X, etc. and what reduction of these means, but how much do those abbreviations mean to an immunologist or biomedical? On the other hand, many immunological abbreviations may not be readily understood by a clinical dermatologist, although clear for i.e. immunologists.

Thank you very much for the comment. As you pointed out, clinical dermatologists are familiar with EASI and SCORAD, but immunologists may not be. Based on this, we cited their respective definitions and described them. We have also added more detail in the part about the JAK-STAT pathway and aryl hydrocarbon receptor (AHR) to aid understanding from an immunological perspective.

Tables should also be explained in full and be understood without having to refer to the text. All abbreviations used in tables also need to be written in full as table footnotes or legend.

Thank you very much for the comment. We added figure legend of table 1 as follow:

Tralokinumab and lebrikizumab are monoclonal antibodies that antagonizes IL-13. CM310 and CBP201 are antibodies binding to a different region of IL-4Rα than that bound by dupilumab. Fezakinumab is a monoclonal antibody against interleukin-22. Nemolizumab is an anti-IL-31 receptor A monoclonal antibody approved in Japan. KPL-716 is an anti-oncostatin M receptor (OSMR) beta antibody that blocks IL-31 and oncostatin M signaling. Spesolimab is an IL-36 receptor antibody for the treatment of generalized pustular psoriasis. CM326 and BSI-045B are monoclonal antibodies, targeting thymic stromal lymphopoietin (TSLP). Delgocitinib, ruxocitinib, tofacitinib, and cerdulatinib are topical JAK inhibitors. These JAK inhibitors prevent from activation of signal transduction and activator of transcription (STAT), a downstream signaling of JAK. Telazorlimab and rocatinlimab are antibodies against OX40. KY1005 is an antibody against OX40L. LEO152020 is an antagonist of histamine H4 receptor (H4R). Tapinarof is a therapeutic aryl hydrocarbon receptor (AHR)-modulating agent and utilized as a topical cream. Crisaborole, difamilast and roflumilast are topical agents of phosphodiesterase (PDE)-4 inhibitor. ShA9 and B244 are bacteriums correcting dysbiosis of AD.

Suggestion: It may also be of interest to mention potential side effects when targeting a certain pathway.

Thank you very much for the comment. We have described adverse events of JAK inhibitors in the part on the Janus kinase (JAK)-signal transduction and activator of transcription (STAT) pathway as follows:

(Lines 217-227)

The JAK-STAT pathway mediates both inflammation and normal physiological functions such as cell proliferation and hematopoietic function [52]. Therefore, adverse event (AE) monitoring is important in treatment with oral JAK inhibitors. The major AEs of oral JAK inhibitors in AD include gastrointestinal disturbances, nasopharyngitis, and headache, which were reported to occur more frequently than in a placebo group [53]. Meanwhile, no cases of malignancy or death were reported [53]. Elevated creatine phosphokinase levels were noted with baricitinib and upadacitinib treatments. Patients with elevated creatine phosphokinase levels were asymptomatic [53], but further investigation is needed. Serious infections, herpes zoster, and acne have been reported to be more common in patients treated with JAK inhibitors, but their frequencies are still very low [44,48,50].

Not explained in full when first mentioned (both abstract, text and table/figures): JAK, PDE4, TSLP, JAK-STAT, SCORAD, EASI, EASI-75, TYK...

Thank you very much for the comment. We have added full spelling for the first occurrence of abbreviations.

Round 2

Reviewer 1 Report

Accept in present form

Manuscript is suitable for publication in my opinion.